# Analysis, Modeling, and Simulation of Thin-Film Cells-Based Photovoltaic Generator Combined with Multilayer Thermoelectric Generator

**DOI:** 10.3390/mi12111342

**Published:** 2021-10-31

**Authors:** Yasir Musa Dakwar, Simon Lineykin, Moshe Sitbon

**Affiliations:** 1Department of Electrical and Electronics Engineering, Ariel University of Samaria, Ariel 40700, Israel; yasirda@ariel.ac.il; 2Department of Mechanical Engineering and Mechatronics, Ariel University of Samaria, Ariel 40700, Israel; simonl@ariel.ac.il

**Keywords:** PVG, TEG, a-Si

## Abstract

A new model for a multi-stage thermoelectric generator (TEG) is developed. An electrical and thermal model is built and simulated for different configurations of photovoltaic (PV) stand-alone hybrid systems, combining different stages of a TEG. The approach is evaluated with and without cooling by coupling a cold plate to a multi-stage hybrid PVTEG system. The model can be adjusted by sizing and specifying the influence of stage number on the overall produced power. Amorphous silicon thin-film (a-Si) is less affected by rising temperature compared to other technology. Hence, it was chosen for evaluating the lower limit gain in a hybrid system under various ambient temperatures and irradiances. The dynamics of the PVTEG system are presented under different coolant water flow rates. Finally, comparative electrical efficiency in reference to PV stand-alone was found to be 99.2% for PVTEG without cooling, 113.5% for PVTEG, and 117.3% for multi-stage PVTEG, accordingly installing multi-stage PVTEG at Israel in a typical year with an average PV yield of 1750 kWh/kW/year generates an extra 24 kWh/year per module hence avoiding fossil energy and equivalent ${\mathrm{CO}}_{2}$ emissions.

## 1. Introduction

According to the International Energy Agency (IEA) [1], the total 2020 worldwide PV power generation capacity is evaluated at 623.2 GW. Enormous 112 GW PV power systems were installed from the start of 2016 up to 2019, with PV power systems holding significant growth potential and having doubled their production. PV generation systems that occupy an essential role for various renewable energy sources such as wind and hydropower hold the most promising solution to the problem of toxic gas emissions resulting from fossil fuel exploitation. Fossil fuels are the main contributor to the increase in greenhouse gases, the most important of which is carbon dioxide, resulting from the combustion of billions of tons of fuel—whether from industrial facilities, power stations, or means of transportation. In contrast, PV power saves 700 million tons of ${\mathrm{CO}}_{2}$ every year. The US energy information administration estimated that world energy consumption in the next five years [2] will be approximately 186,686.27 TWh, with a world average of 2334 h of sun a year and the general efficiency of a standard PV system is 5–20%. Consequently, to meet the annual global demand, an area roughly equal to twice the size of the UK is required to overcome this obstacle. Thus, the production capacity of PV systems must be increased, either through improving efficiency or increasing annual sun hours. As we have an inability to control the latter, it is imperative to improve the efficiency of photovoltaic modules that are connected in a series and parallel to form photovoltaic systems. The main component of the module is the photovoltaic cells [3,4]. A conventional solar cell structure is based on a simple diode p-n semiconductor junction, operating under solar irradiance, containing photons in the range of 0–4 eV. Three cases can occur in the absorption layer depending on the photon energy $E_{\mathrm{ph}}$ relative to the energy band gap $E_{\mathrm{gap}}$:
$E_{\mathrm{ph}}=E_{\mathrm{gap}}$ an electron-hole pair is generated separated by the built-in contact potential of the p–n junction and collected by the metal contacts and produces electric power.$E_{\mathrm{ph}}<E_{\mathrm{gap}}$ photons will propagate through the absorption layer of the solar cell.$E_{\mathrm{ph}}>E_{\mathrm{gap}}$ also absorbed, nevertheless the electron-hole pair occupies a high state in the conduction and valance bands as carriers that tend to occupy a lower energy state close to conduction and valance bands; they release the excess energy that will heat up the lattice in a thermal relaxation process, Boltzmann losses also contribute in heating the lattice.

Since a significant part of the solar spectrum is dissipated as heat significantly reduces the PV’s produced power, several types of research [5,6,7,8,9,10,11,12] were conducted to extract electrical energy through waste heat energy recovery. A thermoelectric generator (TEG) can convert wasted thermal energy into electrical energy through the Seebeck effect when the hot and cold sides hold a temperature gradient.

There are different hybrid PV-TEG configurations: Dianhong Li [5] has studied the impact of concentration ratios on the total efficiency of a system constructed with varying cells of PV and TEG. The TEG is directly attached to the backside of the solar cell and cold plate to cool down the TEG cold side. The results showed an increase in the system’s total electric efficiency with higher concentration ratios. Ershuai Yin [6] compared the influence of different cooling systems (natural air, forced air, and water cooling) with varying cells of PV, and demonstrated the results of hybrid PV TEG system total efficiency. Finally, he concluded that water cooling is the best cooling method to enhance the system’s performance, although these results do not include polymer PV with low concentration ratios. Yuekuan Zhou [7,8,9] developed a numerical model for alternative PV cooling techniques by integrating with phase change materials (PCM) and investigating the performances of PV/T-PCM systems under different inlet water temperatures and various flow rates. The artificial neural network used in order to characterize the optimization function, geometrical and operating parameters were determined using multivariable optimizations to maximize the overall power generation. Adham Makki [10] investigated a PVTEG system with a heat pipe to transfer the heat generation in the PV by vaporizing the inside liquid in the heat pipe and condensing it on the other side, where the TEG was mounted. This configuration reduced the PV cell temperature and increased the total efficiency by 1.5%. Belkacem Zouak [11] proposed a configuration for cooling down the PV cell, where a thermoelectric cooler is used as a heat pump, transferring the heat from the rear surface of the cell by Peltier effect to the TEG. That produces additional power from the extracted heat by Seebeck effect with increasing of total electric power. Esam Elsarrag [12] split the irradiance spectrum with a mirror, which was placed at 45° degrees to the PV cell and the TEG. A wavelength less than 800 nm was reflected towards the PV cell for increasing generation of electric power, whereas the metal layer absorbed the remaining spectrum to produce heat at the hot side of the TEG. The hybrid system has shown an improvement result of 120% in performance over the standard system.

The aim and novelty of this paper:
Create a new and accurate simulation that includes all the hybrid system components that simulatesthe operation of the system electrically and thermally.Enhance the performance of the hybrid system by adding additional TEG stages.Discover the lower limit of power gain from the hybrid system as a-Si is less affected by heating so upgrading solar cells from other technologies with the cooling system and TEG gives further profits.

## 2. System Description and Problem Identification

The standard PV TEG system is represented in Figure 1a. In this system, the TEG is attached directly to the rear surface of the PV module and the cold plate, which is used to increase the temperature gradient across the TEG. In Figure 1b a power flow diagram is illustrated. As the PV absorbs 85% of the incident irradiance while converting 5–20% into electric power depending on the solar cell technology, the rest of the irradiance gets converted into heat and increases the cell temperature. Consequently, there is a decrease in the band gap of a semiconductor; therefore, lower energy is needed to break the bonds. Hence, more photons will be utilized in creating an electron-hole pair in the p–n junction, which slightly increases the PV photon current. The open-circuit voltage is strongly dependent on the reverse saturation current and is highly sensitive to temperature changes. In [13], an equation is derived that describes the negative impact of temperature on the open-circuit voltage. The increase in photon current cannot compensate for the high decrement in the open-circuit voltage of the PV. The final result causes a significant performance drop of the PV. Hence, cooling the solar cell by passing the heat flux through a TEG into the circulating water pipes of the cold plate could enhance the electric power. This process creates a temperature difference across the TEG as the charge carrier in the semiconductor tends to diffuse to lower temperatures, causing an electrical potential to develop across the TEG terminals. Therefore, connecting an external load enables the current to flow and produce extra electric power. It is important to note that TEG efficiency is limited by Carnot efficiency and depends on a figure of merit (zT). It takes into account the electric and thermal conductivity, Seebeck coefficient and examines if the semiconductor material is suitable for power generation. Different materials have different zT according to the range of temperature operation [14]. Amorphous silicon thin-film (a-Si) solar cells are less affected by high temperatures compared to other technologies. It is selected to justify the enhancement in power generation and to acquire the lower limit gain in the efficacy of the PV-TEG combination. Bismuth Tellurium (BiTe) is optimally integrated into the hybrid system due to its high figure of merit in temperatures ranging from 20–100 °C [15,16]. Hence, it constitutes a good match for PV operating temperatures. To validate the hybrid system’s functioning, control the temperatures, and obtain the maximum power point, all the components of the system PV, TEG, and the cold plate should be modeled appropriately and simulated thermally and electrically to achieve a high-quality design with comparison to stand-alone PV.

## 3. Model Development of Photovoltaic, Thermoelectric Generator, and Thermal System Modules

An equivalent electric circuit [17,18] that describes the PV module’s electric behavior contains four components, as shown in Figure 2. A single diode is connected in parallel to a current source $I_{\mathrm{ph}}$ generated due to the photovoltaic effect, while equivalent series resistance $R_{s}$ represents the ohmic collector resistance. Parallel resistance $R_{\mathrm{sh}}$ represents defects in the semiconductor materials, and the output current and voltage $I_{\mathrm{pv}},\;V$ is respectively given by Equation (1) where the second term is the diode current:(1)$$I_{\mathrm{pv}}=N_{P}I_{\mathrm{ph}}-N_{P}I_{0}\left(e^{\frac{V+{\mathrm{IR}}_{s}\left(\frac{N_{s}}{N_{p}}\right)}{{\mathrm{nN}}_{s}V_{T}}}-1\right)-\frac{V+{\mathrm{IR}}_{s}\left(\frac{N_{s}}{N_{p}}\right)}{R_{\mathrm{sh}}}$$
(2)$$I_{\mathrm{ph}}=\left(I_{\mathrm{sc},\mathrm{STC}}+K_{I}\Delta \mathrm{Tsc}\right)\frac{G}{G_{\mathrm{STC}}}$$
(3)$$I_{0}=\frac{I_{\mathrm{sc},\mathrm{STC}}+K_{I}\Delta T}{e^{\left(\frac{V_{\mathrm{oc},\mathrm{STC}}+K_{v}\Delta \mathrm{Tsc}}{{\mathrm{nV}}_{T}}\right)}-1}$$
(4)$$V_{T}=\frac{N_{s}{\mathrm{kBT}}_{\mathrm{pv}}}{q}\;\;\;$$

$N_{P},N_{s}$ represent the number of cells in parallel and series, respectively; $K_{I},K_{v}$ represent the short circuit current open-circuit voltage/temperature coefficients, respectively; and $n,I_{0},V_{T}$ represent the diode ideality factor, the reverse saturation current, and thermal voltage, respectively. $\mathrm{kB},T_{\mathrm{pv}},q$ represents the Boltzmann constant, cell temperature, and electron charge, respectively. $I_{\mathrm{sc},\mathrm{STC}},V_{\mathrm{oc},\mathrm{STC}},G_{\mathrm{STC}}$ represent short circuit current, open-circuit voltage, and irradiance at the STC standard test condition, respectively (Table 1). ΔTsc,G represents the difference between actual cell temperature and STC temperature and actual irradiance, respectively. The $I_{\mathrm{ph}}$ rises as the cell temperature increases, as described in Equation (2). On the other hand, there is a significant drop in the output voltage V, and, as a result, the output power decreases. Hence, The efficiency will drop due to temperature increase, With the following equations describing the negative correlation:(5)$$P_{\mathrm{pv}}={\mathrm{GA}}_{\mathrm{pv}}ɳ$$
(6)$$ɳ={ɳ}_{\mathrm{STC}}\left(1-K_{ɳ}\Delta \mathrm{Tsc}\right)$$

$P_{\mathrm{pv}},A_{\mathrm{pv}},ɳ$ represent the output power, area, and efficiency of the PV module, respectively, and ${ɳ}_{\mathrm{STC}},K_{ɳ}$ represent efficiency at STC and the efficiency temperature coefficient, respectively.

TEGs are direct energy conversion devices that convert heat into electric energy [19]. Once temperature differences exist between the junctions, they lead to a charge in the carrier’s motion according to the Seebeck effect. Joule heating and heat conductance are essential effects to analyze in the TEG operation. Joule heating is caused by current flow and ohmic losses, while heat conduction is the rate of heat transfer through the material and is proportional to temperature difference. According to Fourier’s Law, these processes are given by the energy balance equation at the hot and cold side of the TEG: (7)$$Q_{H}=N_{\mathrm{TEG}}\left({\alpha I}_{\mathrm{TEG}}T_{H}-\frac{1}{2}R_{\mathrm{int}}I_{\mathrm{TEG}}{}^{2}+K_{\mathrm{TEG}}\left(T_{H}-T_{C}\right)\right)$$
(8)$$Q_{C}=N_{\mathrm{TEG}}\left({\alpha I}_{\mathrm{TEG}}T_{C}+\frac{1}{2}R_{\mathrm{int}}I_{\mathrm{TEG}}{}^{2}+K_{\mathrm{TEG}}\left(T_{H}-T_{C}\right)\right)$$

$Q_{H},Q_{C},T_{H},T_{C}$ represent heat energy and temperature at the hot and cold side, respectively, and $N_{\mathrm{TEG}},{\;I}_{\mathrm{TEG}},R_{\mathrm{int}},K_{\mathrm{TEG}}$ represent the number of TEGs, current flow, internal resistance, and thermal conductivity of the TEG, respectively, while α is the Seebeck coefficient. An array of series-connected TEGs can be modeled as an electric circuit with a voltage source and internal resistance connected in a series [20] (see Figure 3), while the output voltage $V_{\mathrm{out}}$, can be calculated by:(9)$${\;V}_{\mathrm{out}}=V_{\mathrm{oc},\mathrm{TEG}}-N_{\mathrm{TEG}}\left(R_{\mathrm{int}}I_{\mathrm{TEG}}\right)$$
(10)$$V_{\mathrm{oc},\mathrm{TEG}}=N_{\mathrm{TEG}}\alpha \left(T_{H}-T_{C}\right)$$
(11)$$I_{\mathrm{TEG}}=\frac{V_{\mathrm{oc},\mathrm{TEG}}}{N_{\mathrm{TEG}}R_{\mathrm{int}}+R_{\mathrm{load}}}$$
(12)$$P_{\mathrm{TEG}}=V_{\mathrm{out}}I_{\mathrm{TEG}}$$

$V_{\mathrm{oc},\mathrm{TEG}}$ is the open-circuit voltage of the array and is directly proportional to the temperature difference and $P_{\mathrm{TEG}},R_{\mathrm{load}}$ is the output power and load resistance, respectively

As the output power of the PV TEG system strongly depends on the working temperature of the system, a one-dimensional module for heat transfer is proposed (see Figure 4). By solving a system of energy balance equations, we can obtain the desired temperature of the PV module and the TEG Hence the output power of the hybrid system. Standard photovoltaic modules contains six layers [21]: covering glass, anti-reflecting coating (ARC), solar cell or thin-film, Ethylene Vinyl Acetate (EVA) layer, metal sheet, and back cover. The covering glass has a high transmittance and protects the PV cells from external damages. ARC provides a path for photons into the solar cell. Some photovoltaic modules use wafers or thin-film depending on the technology used in the manufacturing process. EVA for encapsulation of solar cells with a covering glass in the rear side of the solar cell/thin-film, metal contact (gold/silver/aluminum) is used by the screen printing process to collect the carries in the back side of the solar cell. The back cover is made of tempered black glass and used for insulation. The TEG is constructed by two ceramic plates to equally dissipate the heat across the semiconductor’s legs and very thin copper strips that connect the p–n pairs and provide a flow path for the current. The energy balance equations for the various layers of the hybrid PV TEG are given below.

PV cells:(13)$$\rho_{\mathrm{pv}}v_{\mathrm{pv}}C_{\mathrm{pv}}\frac{{\mathrm{dT}}_{\mathrm{pv}}}{\mathrm{dt}}=\tau_{g}{\mathrm{GA}}_{\mathrm{pv}}-U_{\mathrm{pv}-g}\left(T_{\mathrm{pv}}-T_{g}\right)-U_{\mathrm{pv}-\mathrm{EVA}}\left(T_{\mathrm{pv}}-T_{\mathrm{EVA}}\right)-P_{\mathrm{pv}}\;\;\;\;\;$$

Front glass:(14)$$\rho_{g}v_{g}C_{g}\frac{{\mathrm{dT}}_{\mathrm{pv}}}{\mathrm{dt}}=U_{\mathrm{pv}-g}\left(T_{\mathrm{pv}}-T_{g}\right)-h_{g-a}\left(T_{g}-T_{\mathrm{amb}}\right)\;\;\;\;\;$$

EVA:(15)$$\rho_{\mathrm{EVA}}v_{\mathrm{EVA}}C_{\mathrm{EVA}}\frac{{\mathrm{dT}}_{\mathrm{EVA}}}{\mathrm{dt}}=U_{\mathrm{pv}-\mathrm{EVA}}\left(T_{\mathrm{pv}}-T_{\mathrm{EVA}}\right)-U_{\mathrm{EVA}-\mathrm{rm}}\left(T_{\mathrm{EVA}}-T_{\mathrm{rm}}\right)$$

Rear contact:(16)$$\rho_{\mathrm{rm}}v_{\mathrm{rm}}C_{\mathrm{rm}}\frac{{\mathrm{dT}}_{\mathrm{rm}}}{\mathrm{dt}}=U_{\mathrm{EVA}-\mathrm{rm}}\left(T_{\mathrm{EVA}}-T_{\mathrm{rm}}\right)-U_{\mathrm{rm}-b}\left(T_{\mathrm{rm}}-T_{b}\right)$$

Back cover:(17)$$\rho_{b}v_{b}C_{b}\frac{{\mathrm{dT}}_{b}}{\mathrm{dt}}=U_{\mathrm{rm}-b}\left(T_{\mathrm{rm}}-T_{b}\right)-U_{b-\mathrm{cr}}\left(T_{b}-T_{\mathrm{cr}}\right)\;\;\;\;$$

Ceramic hot side:(18)$$\rho_{\mathrm{cr}}v_{\mathrm{cr}}C_{\mathrm{cr}}\frac{{\mathrm{dT}}_{H}}{\mathrm{dt}}=U_{b-\mathrm{cr}}\left(T_{b}-T_{H}\right)-N_{\mathrm{TEG}}\left({\alpha I}_{\mathrm{TEG}}T_{H}-\frac{1}{2}R_{\mathrm{int}}I_{\mathrm{TEG}}{}^{2}+K_{\mathrm{TEG}}\left(T_{H}-T_{C}\right)\right)$$

Ceramic cold side:(19)$$\rho_{\mathrm{cr}}v_{\mathrm{cr}}C_{\mathrm{cr}}\frac{{\mathrm{dT}}_{C}}{\mathrm{dt}}=N_{\mathrm{TEG}}\left({\alpha I}_{\mathrm{TEG}}T_{C}+\frac{1}{2}R_{\mathrm{int}}I_{\mathrm{TEG}}{}^{2}+K_{\mathrm{TEG}}\left(T_{H}-T_{C}\right)\right)-U_{\mathrm{cr}-\mathrm{cp}}\left(T_{C}-T_{\mathrm{cp}}\right)$$

Cold plate:(20)$$\rho_{\mathrm{cp}}v_{\mathrm{cp}}C_{\mathrm{cp}}\frac{{\mathrm{dT}}_{\mathrm{cp}}}{\mathrm{dt}}=U_{\mathrm{cr}-\mathrm{cp}}\left(T_{C}-T_{\mathrm{cp}}\right)-h_{\mathrm{cp}-w}\left(T_{\mathrm{cp}}-T_{w}\right)\;\;\;\;\;$$

ρ,v,C,T are the density, volume, specific heat, and temperature of each layer, respectively (Table 2). U is the overall heat transfer coefficient (thermal impedance) that can be calculated according to the following equation: $U=\frac{\mathrm{Ak}}{l}$, where A,k,l are the area, thermal conductivity, and thickness of each layer, respectively. $h_{g-a}=\left(5.7+3.8v\right)\cdot A$ is the overall convection coefficient where v is the air velocity at the front glass layer,${\;\tau }_{g}=0.686375$ is the photovoltaic absorptivity and takes into consideration the packing factor, transmissivity, and reflectivity of the glass. $h_{\mathrm{cp}-w}$ is the overall heat transfer between the surface of the cold plate and the cooling water and depends on various parameters such as diameter, length, and the number of pipes in the cold plate. For the sake of simplicity, a pre-designed commercially available cold plate is used in the hybrid system. In general, the manufacture [22] provides a datasheet containing two graphs, the overall thermal resistance between the surface and the cooling water. The pressure drop across the cold plate is a function of the water flow rate. By multiplying the pressure drop Δp and the flow rate $m_{w}^{\prime }$ it can get the necessary power to drive the water through the pipes. The electric power consumption of a pump is given by:(21)$$P_{\mathrm{pump}}=\frac{\Delta pm_{w}^{\prime }}{{ɳ}_{\mathrm{pump}}}$$

${ɳ}_{\mathrm{pump}}$ is the pump efficiency and evaluated at 90%. The total efficiency of the system is given by:(22)$${ɳ}_{\mathrm{tot}}=\frac{P_{\mathrm{pv}}+P_{\mathrm{TEG}}-P_{\mathrm{pump}}}{{\mathrm{GA}}_{\mathrm{pv}}}$$

According to [23] TEG parameters can be obtained from manufacturer’s data (Table 3) by the following equations:(23)$$\;\alpha =\frac{V_{\max}}{T_{h}}$$
(24)$$R_{\mathrm{int}}=\frac{V_{\max}\left(T_{h}-\Delta T_{\max}\right)}{I_{\max}T_{h}}$$
(25)$${\;K}_{\mathrm{TEG}}=\frac{I_{\max}V_{\max}\left(T_{h}-\Delta T_{\max}\right)}{\Delta T_{\max}2T_{h}}$$

## 4. Methodology and Simulation

The hybrid system was simulated using MATLAB/Simulink according to the block diagram in Figure 5. In the first stage, the temperatures are obtained by solving the thermal partial differential equation (PDE) using system Euler’s method. Subsequently, relevant values are fed into the PV and TEG modules. The output voltage and current are connected into two DC-DC converters, with the duty cycle determined according to the MPPT by P and O Algorithm that keeps tracking the maximum power point and adapts the duty cycle according to changes in the irradiance and temperatures [25]. Consequently, maximum power is extracted from the system at all times.

Running the simulation with $G,T_{\mathrm{amb}},h_{\mathrm{cp}-w}$ which changes as shown in Figure 6, and with the purpose of demonstrating hybrid system performance over the first half of the day, a fixed step size solver is used with $10^{6}$ samples (iterations) every second defining Euler’s step at $10^{-2}$ s and multiplying the number of samples by Euler’s step equals $10^{4}$ s, which is equivalent to two hours and 46.66 min in real-time. Accordingly, simulating for two seconds is sufficient to analyze the system performance in the desired hours of the sunrise period and until reaching steady irradiance and ambient temperatures. Initially, all the system temperatures were entered in intervals around the $T_{\mathrm{amb}}$ value, as the sun rises, irradiance and ambient temperature increase and reach different intensities according to annual seasons of $1000,800,600\;W/m^{2}$ and 50,35,25 °C. The warming process takes about one hour, and the PV will generate electric power and heat until the system reaches the equilibrium point at approximately 50 °C above the ambient temperature. While there is no significant temperature difference across the TEG, a slight power can still be generated. It can be utilized to start cooling the PV module by activating the water pump and circulating the coolant flow across the cold plate. Gradually, it will increase the temperature difference across the TEG ($\Delta T=T_{H}-T_{C}$), hence, increasing $P_{\mathrm{TEG}}$. On the other hand, cooling down the cold side of the TEG will directly cool down the PV module. Consequently, it will cause ΔT to decrease while significantly increasing the output power of $P_{\mathrm{PV}}.$ Due to the inverse influence of the temperature on PV power output, this process continues until the system reaches a new equilibrium point. The example results of the process are shown in Figure 7-(Table 4) for various irradiances G and heat transfer coefficients $h_{\mathrm{cp}-w}=400,800\;W/{}^{\circ}C$. that are associated directly with $m_{w}^{\prime }=0.4,1.1\frac{L}{\min}$ and $P_{\mathrm{pump}}=0.078,0.5\;W$ accordingly, where the red and the blue curves describe $P_{\mathrm{PV}}\;\mathrm{and}\;P_{\mathrm{TEG}},$ respectively.

$P_{\mathrm{pv},\mathrm{imp}}$ is the improved power of PVTEG compared to PV stand alone. The hybrid system PVTEG generates more power than a conventional stand-alone PV system under various irradiances and ambient temperatures, as illustrated in Figure 8 and Figure 9, with assuming constant airspeed of v=0.25 m/s and specific value of water flow $m_{w}^{\prime }=0.4\frac{L}{\min}.$ These assumptions can define and guarantee a constant thermal resistance between the cold plate and the water, $h_{\mathrm{cp}-w}$ at 400 W/ °C. Figure 8 demonstrates the results by a continuous and dashed curve representing the total output power of the PV stand-alone and PVTEG systems, respectively. The various curve colors corresponded to different irradiance intensities with constant $T_{\mathrm{amb}}=25\;{}^{\circ}C$, and clearly show the hybrid system’s advantage compared to a conventional PV system for different radiation intensities, due to the water flow which cools the PVTEG. Consequently, more power can be extracted from the system, and, furthermore, it allows continuous stable output power compared to the PV, even after the temperature and radiation stabilization. The PV power starts to decrease after reaching the peak power due to the part of the irradiance converted into heat and increased the PV panel temperature, a well-known cause of reducing PV output power up to the equilibrium point. The disparities between the systems are noticeable, consequently resulting in a significant difference in efficiency terms. As the incoming energy by irradiation (G) increases, more energy is converted and dispatched as heat in the PV panel. In contrast, the PVTEG removes the extra heat and converts a part of it into usable electric power, maintaining extended and effective stable operation. While both PV stand-alone and PVTEG systems have shown a decreasing tendency with the increase in ambient temperatures, due to negative correlation of temperature increment on the efficiency of solar cells as shown in Figure 9, where the various curve colors represent different ambient temperatures under constant $G=1000\;W/m^{2}$, it is clearly shown that the PVTEG output power is always above the PV stand-alone power.

Although there is an increase in power production in PVTEG systems, and it is more efficient than the stand-alone PV system, a large part of the energy goes through the TEG and heats the cooling water without utilization due to its low efficiency. In order to decrease these losses, additional TEG stages were added between the cold plate and the PV module. Adding multiple stages can utilize additional efficiency from the crossing heat and increase the output power. On the other hand, each stage increases the overall thermal resistance between the PV and the cold plate. It is important to note that the high thermal resistance prevents removing the extra heat from the PV (trade-off). For the purpose of demonstrating the influence of thermal resistance, a series of simulations were extracted with constant $G,h_{\mathrm{cp}-w}\;,v,T_{\mathrm{amb}}$ and various TEG stages were added to examine the maximum power generation possible. According to Figure 10, which describes the effect of installing additional TEGs on total power output and on the PV temperature, 14 stages are the upper threshold for extracting maximum energy. Adding more layers will reduce the power produced by the system. As $T_{\mathrm{amb}}=T_{w}$ and with the increases of the overall thermal resistance between the PV and the cold plate, the heat passes through the small resistance of the top glass layer and dissipates to the environment, causing lower input power to the TEG layers. Likewise, the high thermal resistance between the cold plate and the PV prevents the cooling effect of the water from crossing the layers and influencing the solar cell, due to the increase in the PV temperature. As a result, there is a reduction in the overall power output of the system.

In Figure 11 and Figure 12, a comparison of four configurations is demonstrated: PV stand-alone (blue line), PVTEG without cooling (red line), PVTEG with cooling (green dashed line), and PVTEG with cooling and additional layers (black dashed line). According to the results, PVTEG without cooling holds only a slight advantage over PV stand-alone due to increasing mass, which mitigates the temperature rise in the system. However, after the system reaches an energy balance, there is no significant ΔT across the TEG. The $P_{\mathrm{TEG}}$ is compensated with higher $T_{\mathrm{pv}}$ and lower $P_{\mathrm{PV}}$ without any contribution, and even a small reduction in the system’s total power can be observed throughout the day.

Adding a cold plate across the TEG’s lower layer has a significant impact on power production. Consequently, the system efficiency is 13.56% higher than PV stand-alone. The most substantial advantage of the hybrid system is the operation under low $T_{\mathrm{pv}}$ and higher $P_{\mathrm{PV}}$. Secondly, the increase in ΔT, and hence in $P_{\mathrm{TEG}}$ when adding the optimal number of TEGs 14 stages in number to the system, as conducted from the results of Figure 10. It enhances total power generation and efficiency but requires more time to settle due to augmented mass in the system, as demonstrated graphically in Figure 11 and Figure 12.

## 5. Conclusions

A new multi-stage TEG and amorphous silicon model was developed. The TEGs were installed between a PV and a cold plate. This configuration generated more additional electric power than a conventional stand-alone PV and single PVTEG and increased efficacy. In this study, amorphous silicon (a-Si) solar cells were modeled and simulated. Typically, a-Si holds a low-efficiency temperature coefficient compared to other technology. The results showed an approximate 13% increase in efficiency compared to a stand-alone PV. Consequently, there is some benefit to upgrading PV systems with other technologies. Another augmentation of efficacy can be attained by adding a multi-stage TEG to the system, which can utilize more power from the wasted heat but increases the solar cell temperature. As a-Si is affected less by temperature increase, a maximum number of TEG stages could be implemented, leading to higher output powers. Electrical efficiency of a PV, stand-alone PVTEG, PVTEG without cooling, and multi-stage PVTEG were found to be 5.54%, 5.5%, 6.29%, and 6.5%, respectively. In addition, the multi-stage PVTEG, due to increasing mass, needs more time to reach the equilibrium point.

## 6. Research Limitations, Future Prospects

According to the results in Table 4, activating the cold plate with a higher $m_{w}^{\prime }$ hence $h_{\mathrm{cp}-w}$ does not necessarily increase the power produced by the hybrid system. Thus, an optimal $m_{w}^{\prime }$ should be determined for maximum power generation. Development of the equation that will include and take into account the increase in $T_{w}$, variation in airspeed, affects $h_{g-a}$ as ν increases. Eventually, more heat dissipates to the ambient, which leads to a decrement of the incoming power to TEGs, accordingly, lower $P_{\mathrm{TEG}}$ and also the optimal stages number depended on the magnitude of $h_{g-a}\;.$ All the mentioned problems should be investigated in future works. For example, exploiting the split mirror, concentrating the incoming irradiance and integrating PCM which could lead to higher overall electric power production and efficiency.

## Figures and Tables

**Figure 1 micromachines-12-01342-f001:**
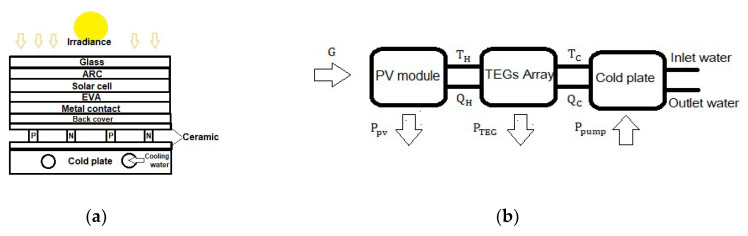
(**a**) Standard PV TEG system configuration, (**b**) Power flow digram.

**Figure 2 micromachines-12-01342-f002:**
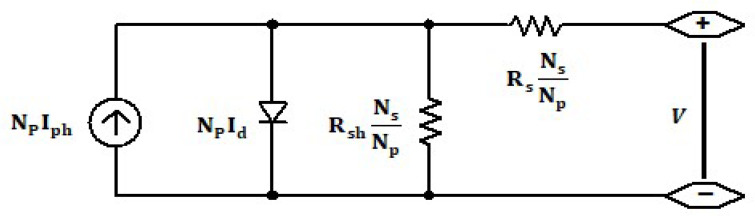
Photovoltaic electric module.

**Figure 3 micromachines-12-01342-f003:**
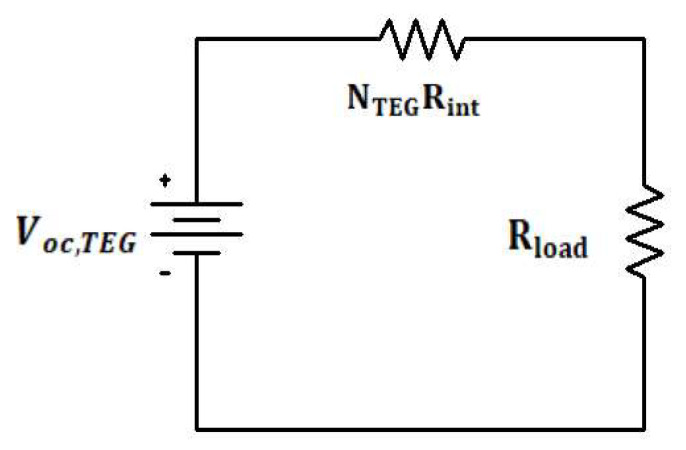
Thermoelectric generator module.

**Figure 4 micromachines-12-01342-f004:**
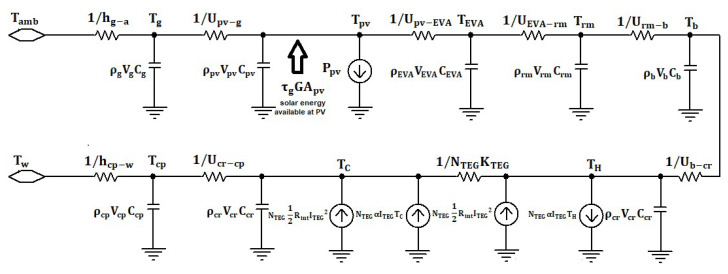
Thermal module of hybrid PVTEG system.

**Figure 5 micromachines-12-01342-f005:**
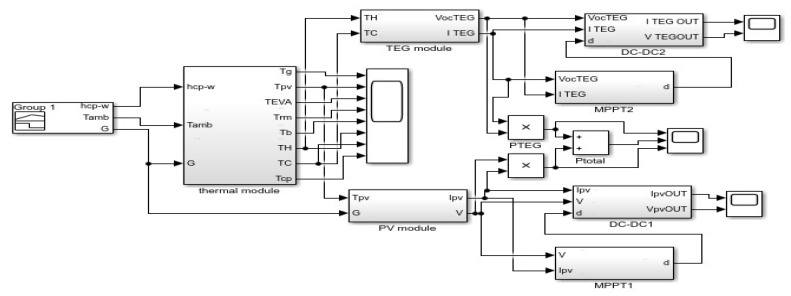
Block diagram PVTEG.

**Figure 6 micromachines-12-01342-f006:**
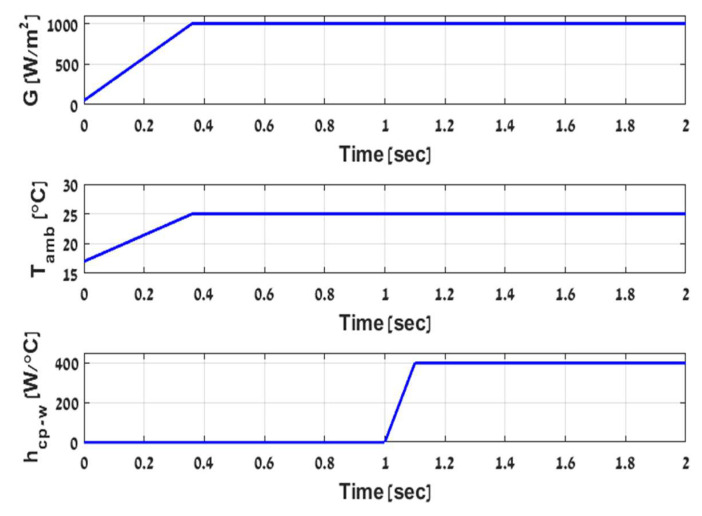
Illustration of simulation input parameters.

**Figure 7 micromachines-12-01342-f007:**
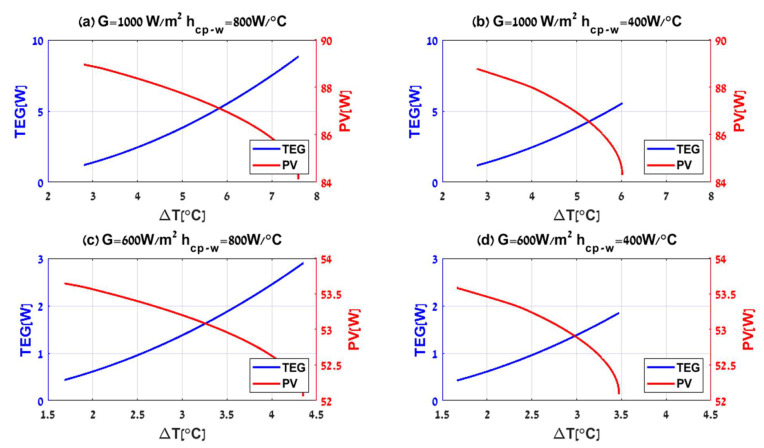
PVTEG dynamics under various irradiances G and water flows $m_{w}^{\prime }$ according to headlines (**a**–**d**).

**Figure 8 micromachines-12-01342-f008:**
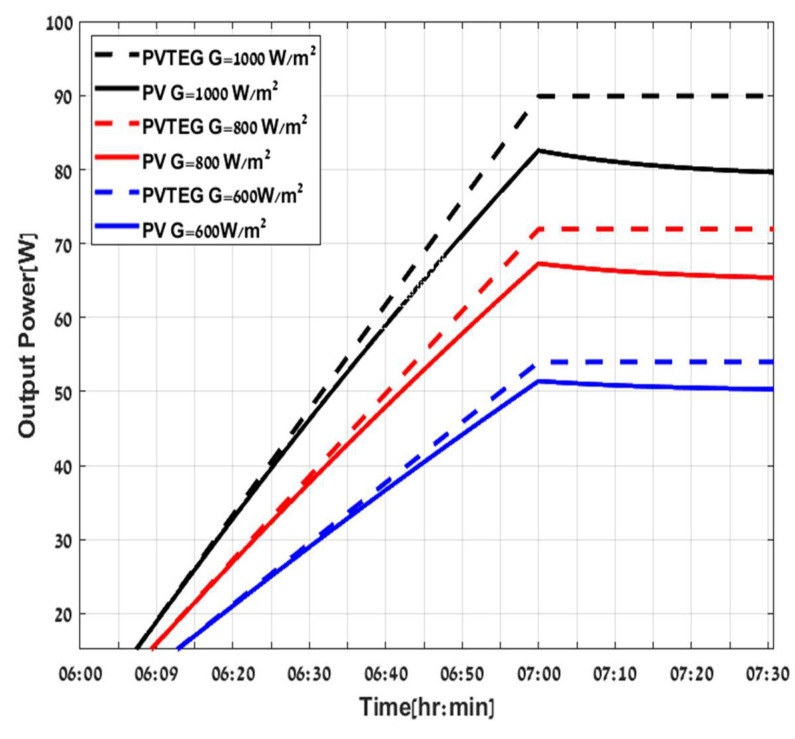
PV (continuous line) PVTEG (dashed line) at different G irradiance.

**Figure 9 micromachines-12-01342-f009:**
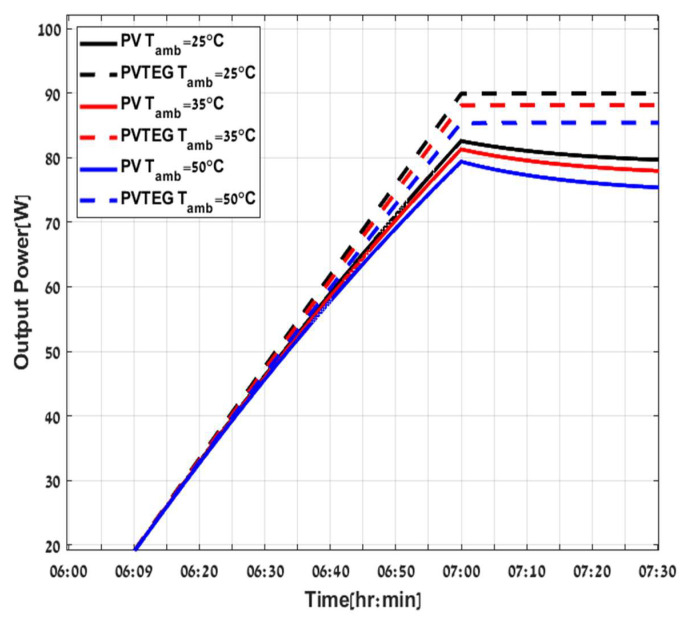
PV (continuous line), PVTEG (dashed line) at different $T_{\mathrm{amb}}$ ambient temperatures.

**Figure 10 micromachines-12-01342-f010:**
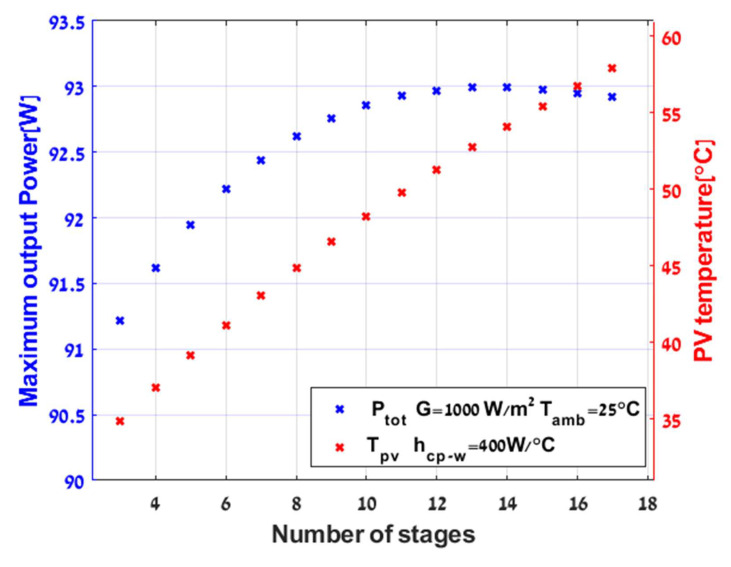
Maximum output power as a function of variable TEG stages.

**Figure 11 micromachines-12-01342-f011:**
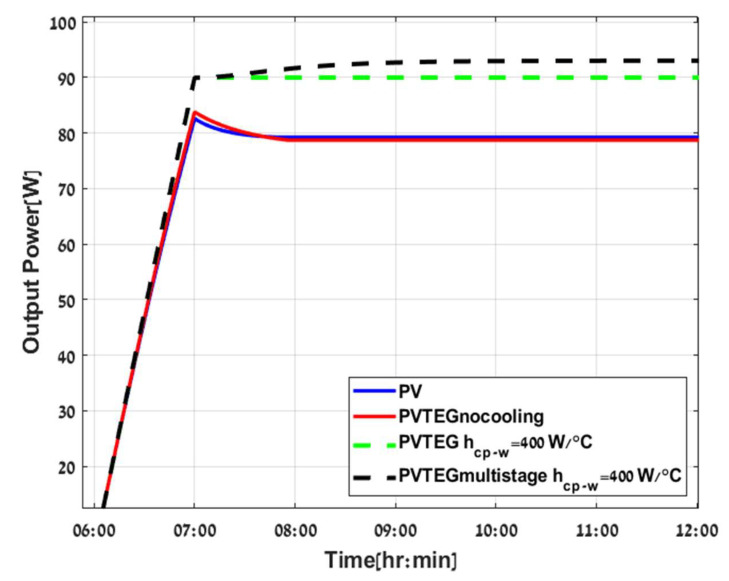
Configuration of PV stand-alone (blue, continuous line), PVTEG without cooling (red, continuous line), PVTEG (green dashed line), and PV Multistage TEG (black dashed line).

**Figure 12 micromachines-12-01342-f012:**
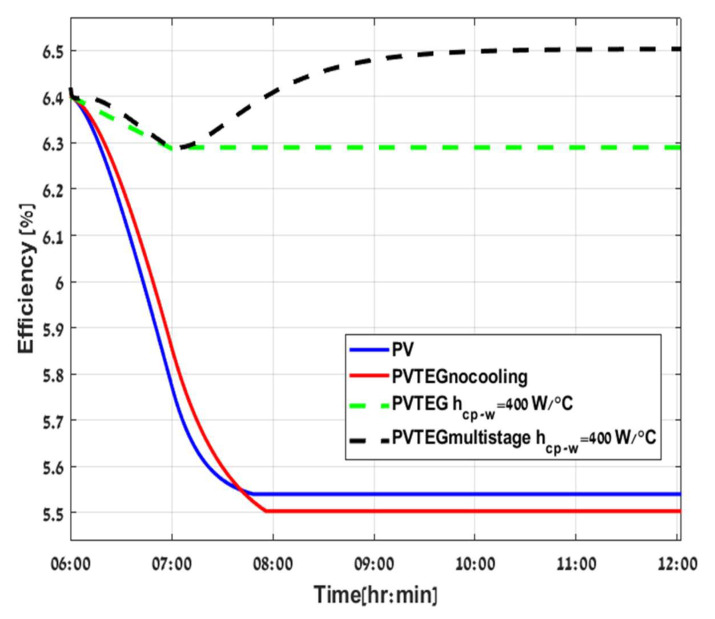
Configuration of PV stand-alone (blue, continuous line), PVTEG without cooling (red, continuous line), PVTEG (green dashed line), and PV Multistage TEG (black dashed line).

**Table 1 micromachines-12-01342-t001:** Parameters from datasheets.

PV	Specification
module	MA100T2.
$I_{\mathrm{sc},\mathrm{STC}}$	1.17 A
$V_{\mathrm{oc},\mathrm{STC}}$	141 V
$G_{\mathrm{STC}}$	$1000\;W/m^{2}$
ɳ	6.3%
$K_{I},$	+0.09%/K
$K_{v}$	−0.33%/K
$K_{ɳ}$	−0.2%/K

**Table 2 micromachines-12-01342-t002:** Design parameters of PV and thermos physical properties of the layer [21].

	Units	Glass	PV	EVA	Rear Contact	Back Cover	Ceramic
ρ	$\mathrm{kg}/m^{3}$	3000	2200	960	2700	3000	2700
C	J/kg K	500	770	2090	900	500	900
k	W/m K	1.8	1.5	0.35	237	1.8	24
l	m	0.0032	0.000015	0.000762	0.00001	0.0032	0.0005
A	$m^{2}$	1.43	1.43	1.43	1.43	1.43	1.43

**Table 3 micromachines-12-01342-t003:** Parameters from datasheets [24].

TEG	Specification
module	TB-71-1.4-3.175
$V_{\max}$	9.1 V
$I_{\max}$	2.9 A
$\Delta T_{\max}$	72 K

**Table 4 micromachines-12-01342-t004:** Results according to Figure 7.

	$P_{pv}\left[W\right]$	ΔT[ °C] (at equilibrium)	$P_{TEG}\left[W\right]$	$P_{pv,imp}\left[W\right]$	${ɳ}_{tot}\left[\%\right]$
Figure 7a	88.95	2.796	1.199	10.26	6.26
Figure 7b	88.78	2.768	1.176	10.5	6.28
Figure 7c	53.64	1.679	0.432	3.292	6.24
Figure 7d	53.58	1.662	0.423	3.645	6.28

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
