# Peer review of "Analysis, Modeling, and Simulation of Thin-Film Cells-Based Photovoltaic Generator Combined with Multilayer Thermoelectric Generator"

_micromachines, 2021, doi:10.3390/mi12111342_

Round 1

Reviewer 1 Report

The work focused on Analysis, Modeling, and Simulation of thin-film cells based photovoltaic generator combined with multilayer thermoelec-tric generator. The manuscript is within the scope of the Journal. In order to help improve the paper quality, my suggestions and comments are shown below.

1) Abstract: add one sentence to highlight the significance of the work at the end of the paper.

2) Nomenclature: suggest to re-order to follow the alphabetical order

3) Introduction: Scientific gaps need to be clearly identified. The reviewer concerns the novelty and contribution of this study. Originality novelty of the work has not been clearly presented. In order to improve the readability and help readers to follow, it is better to firstly list the identified scientific gaps, before introducing the research framework or methodology. Furthermore, it is better to summarise the novelty and contribution at the end of introduction, with point-by-point items.  

PV cooling technologies are better to be reviewed, especially with phase change materials. Some references are listed below and recommended to be cited.

Multi-level uncertainty optimisation on phase change materials integrated renewable systems with hybrid ventilations and active cooling. Energy, 2020. DOI: https://doi.org/10.1016/j.energy.2020.117747

Passive and active phase change materials integrated building energy systems with advanced machine-learning based climate-adaptive designs, intelligent operations, uncertainty-based analysis and optimisations: A state-of-the-art review. Renewable & Sustainable Energy Reviews 2020. DOI: https://doi.org/10.1016/j.rser.2020.109889

A state-of-the-art-review on phase change materials integrated cooling systems for deterministic parametrical analysis, stochastic uncertainty-based design, single and multi-objective optimisations with machine learning applications. Energy and Buildings 2020. DOI: https://doi.org/10.1016/j.enbuild.2020.110013

A review on cooling performance enhancement for phase change materials integrated systems—flexible design and smart control with machine learning applications. Building and Environment 2020. DOI: https://doi.org/10.1016/j.buildenv.2020.106786

Machine learning-based optimal design of a phase change material integrated renewable system with on-site PV, radiative cooling and hybrid ventilations—study of modelling and application in five climatic regions. Energy, 2020. DOI: https://doi.org/10.1016/j.energy.2019.116608

Machine-learning based study on the on-site renewable electrical performance of an optimal hybrid PCMs integrated renewable system with high-level parameters’ uncertainties. Renewable Energy, 2020. DOI: https://doi.org/10.1016/j.renene.2019.11.037

Exergy-based optimisation of a phase change materials integrated hybrid renewable system for active cooling applications using supervised machine learning method. Solar Energy, 2020, 195: 514-526.

Zhou, Y., Zheng, S., Zhang, G. Artificial neural network based multivariable optimization of a hybrid system integrated with phase change materials, active cooling and hybrid ventilations. Energy Conversion and Management. 2019. DOI: https://doi.org/10.1016/j.enconman.2019.111859.

Zhou, S. Zheng, G. Zhang. Study on the energy performance enhancement of a new PCMs integrated hybrid system with the active cooling and hybrid ventilations. Energy. 179 (2019) 111-128.

4) Section titles need to be checked to avoid the repetition, such as 3. Photovoltaic module and 3. Thermoelectric generator module. Moreover, it is better to put Section 3, 3, and 4 in one section, because they belong to model development.

5) By activating the water pump and circulating the coolant flow across the cold plate, the cold side of the TEG will directly cool down the PV module. The reviewer is interested in the magnitude of power consumption in pumps and improved power from PV module.

6) Quality of figures needs to be improved. Moreover, the y-axis in Fig. 7 needs to be keep the same, for the comparison purpose.

7) As the proposed system is an integrated system with amorphous silicon (a-Si) solar cells, TEGs and active water based cooling. Specifications on energy performance of each subsystem will be useful to help readers to follow and get in-depth understanding.

8) One separate section is suggested to be added, to describe Research limitations, challenges and future prospects.

Overall, this manuscript is well-written, organised and described. The reviewer suggests the major revision.

Author Response

Dear Editor,

First, we would like to thank You and the reviewers for the valuable and challenging comments. Please see below the comments (black) and the authors’ responses (red). Significant changes are also highlighted in red in the revised paper. The manuscript has undergone a thorough revision, intended to improve its clarity and readability.

Best Regards,

The authors.

Reviewer 1 Comments to the Author:

The work focused on Analysis, Modeling, and Simulation of thin-film cells based photovoltaic generator combined with multilayer thermoelec-tric generator. The manuscript is within the scope of the Journal. In order to help improve the paper quality, my suggestions and comments are shown below.

- Thank You for the review

  • Abstract: add one sentence to highlight the significance of the work at the end of the paper.
  • Line 34-37, estimates the average extra energy per year per module and reducing fossil fuel consumption.

  • Nomenclature: suggest to re-order to follow the alphabetical order
  • Line 47, all Nomenclature reorganized and a new one assigned to the volume and air velocity to avoid repetition.

  • Introduction: Scientific gaps need to be clearly identified. The reviewer concerns the novelty and contribution of this study. Originality novelty of the work has not been clearly presented. In order to improve the readability and help readers to follow, it is better to firstly list the identified scientific gaps, before introducing the research framework or methodology. Furthermore, it is better to summarize the novelty and contribution at the end of introduction, with point-by-point items.
  • Line 82, the main scientific gap presented.
  • Line 113, presenting the novelty at the end of the introduction, with point-by-point items.
  • Line 95, PV cooling using PCM has been introduced and added to conventional cooling technologies.

  • Section titles need to be checked to avoid the repetition, such as 3. Photovoltaic module and 3. Thermoelectric generator module. Moreover, it is better to put Section 3, 3, and 4 in one section, because they belong to model development.
  • Line 166, all the mentioned sections gather in section 3 under model development.

  • By activating the water pump and circulating the coolant flow across the cold plate, the cold side of the TEG will directly cool down the PV module. The reviewer is interested in the magnitude of power consumption in pumps and improved power from PV module.
  • Line 320, presents the magnitude of power consumption of the pumps.
  • Quality of figures needs to be improved. Moreover, the y-axis in Fig. 7 needs to be keep the same, for the comparison purpose.
  • All the figures improved using bold axis and accurate data to highlight the differences between the systems.
  • As the proposed system is an integrated system with amorphous silicon (a-Si) solar cells, TEGs and active water based cooling. Specifications on energy performance of each subsystem will be useful to help readers to follow and get in-depth understanding.
  • Line 326, all the subsystem components power instead of energy generation presented in table 4
  • One separate section is suggested to be added, to describe Research limitations, challenges and future prospects.
  • Line 418, a separate section which describe research limitations, challenges, and future prospects.
  • More references are added as suggested

Overall, this manuscript is well-written, organised and described. The reviewer suggests the major revision.

  • Thank You

Reviewer 2 Report

The essay contains interesting and new elements, but there are a number of editing errors that detract from the overall picture.

The notation system is complex and difficult to use. Chapter numbering needs correction.

Line 22: C specific heat capacity J/kgK to be written in kg with small k, v- velocity air speed is enough to be included once in the velocity expression, the explanations are not vertically aligned

Line 48-49-50: paragraphs are not vertically aligned

Line 126-129: the formulas slip into each other

Line 141: Hence is in lower case, between sentences

Line 148: This is now chapter 4

Line 246: Figure 6 the captions need to be corrected, the indexes are not the right size. It is not clear whether the 4 seconds is the actual time or the simulated time.

Line 271: Figures 7 to 12. Many editing errors, captions slip into each other, letters in brackets stick out

Line 299: letters and numbers and other characters are not aligned horizontally

Author Response

Dear Editor,

First, we would like to thank You and the reviewers for the valuable and challenging comments. Please see below the comments (black) and the authors’ responses (red). Significant changes are also highlighted in red in the revised paper. The manuscript has undergone a thorough revision, intended to improve its clarity and readability.

Best Regards,

The authors.

Reviewer 2 Comments to the Author:

The essay contains interesting and new elements, but there are a number of editing errors that detract from the overall picture.

  • Thank You for the review.

The notation system is complex and difficult to use. Chapter numbering needs correction.

  • Line 166 all the mentioned sections gather in section 3 under model development

Line 22: C specific heat capacity J/kgK to be written in kg with small k, v- velocity air speed is enough to be included once in the velocity expression, the explanations are not vertically aligned

  • Checked and corrected.

Line 48-49-50: paragraphs are not vertically aligned

Line 126-129: the formulas slip into each other

Line 141: Hence is in lower case, between sentences

  • Checked and corrected.

Line 148: This is now chapter 4

  • Checked and corrected.

Line 246: Figure 6 the captions need to be corrected, the indexes are not the right size. It is not clear whether the 4 seconds is the actual time or the simulated time.

  • All the figures improved using bold axis and accurate data to highlight the differences between the systems.

Line 271: Figures 7 to 12. Many editing errors, captions slip into each other, letters in brackets stick out

  • All the figures improved using bold axis and accurate data to highlight the differences between the systems.

Line 299: letters and numbers and other characters are not aligned horizontally

  • Line 308  Checked  and corrected

Reviewer 3 Report

The manuscript needs further development. The following recent papers may provide a further analysis in the insight of the photon to charge conversion in a-Si photon and electron interactions  in the introduction.

1. Energies 202114(11), 3022; doi:10.3390/en14113022

The design of the experiment and the model is suggested to be more analytical and the description of the MATLAB / Simulink algorithm has to be better explain (See the following ref.2)

2. Salman, S., AI, X. & WU, Z. Design of a P-&-O algorithm based MPPT charge controller for a stand-alone 200W PV system. Prot Control Mod Power Syst 3, 25 (2018). https://doi.org/10.1186/s41601-018-0099-8

The model lacks of a figure and experimental analysis to explain how the data have been derived from.

A deeper insight of the results may be helpful for the readers. E.g. try to connect the single and multi stage analysis with the physical layers interactions of the system (ref.1)

Citations are missing of the narrative to clarify the description of the process and the references has to be enriched with recent ones.

Moreover the authors need to answer to the following questions:

What is the new body of knowledge in the system design and implementation

What is the boundaries of the system

What is the limitations and further research development.

Clear and accurate performance of the system components is required

Author Response

Dear Editor,

First, we would like to thank You and the reviewers for the valuable and challenging comments. Please see below the comments (black) and the authors’ responses (red). Significant changes are also highlighted in red in the revised paper. The manuscript has undergone a thorough revision, intended to improve its clarity and readability.

Best Regards,

The authors.

Reviewer 3 Comments to the Author:

The manuscript needs further development. The following recent papers may provide a further analysis in the insight of the photon to charge conversion in a-Si photon and electron interactions  in the introduction.

  • Thank You for the review.

  1. Energies202114(11), 3022; doi:10.3390/en14113022
  • Line 70, the mentioned paper, has been added to the introduction about photon to charge conversion

The design of the experiment and the model is suggested to be more analytical and the description of the MATLAB / Simulink algorithm has to be better explain (See the following ref.2)

  1. Salman, S., AI, X. & WU, Z. Design of a P-&-O algorithm based MPPT charge controller for a stand-alone 200W PV system. Prot Control Mod Power Syst3, 25 (2018). https://doi.org/10.1186/s41601-018-0099-8
  • Line 294,  the mentioned paper, has been cited and  to provide explain to MPPT

The model lacks of a figure and experimental analysis to explain how the data have been derived from.

  • All the derived data are according to the input parameters in Fig. 6
  • Each PDE in heat transfer equation was solved by integration with the Euler method and feed into  in the simple TEG and PV  model, which can be easily built using Simulink.

A deeper insight of the results may be helpful for the readers. E.g. try to connect the single and multi stage analysis with the physical layers interactions of the system (ref.1)

Citations are missing of the narrative to clarify the description of the process and the references has to be enriched with recent ones.

  • corrected

Moreover the authors need to answer to the following questions:

What is the new body of knowledge in the system design and implementation

  • Line 113 presenting the novelty at the end of the introduction, with point-by-point items. 

What is the limitations and further research development.

  • Line 418 separate section describing  Research limitations, challenges, and future prospects.

Clear and accurate performance of the system components is required

  • All the PDE and data can also be fed into matlab and solved by "fsolve" to guarantee accurate results. Also, the results have been compared to TEG module, which D.r Simon Linyken developed and published in several types of research. All the previous methods provide the same results

Round 2

Reviewer 1 Report

The manuscript is well revised and all comments from reviewer are addressed.